# Effect of Medium Pressure Ultraviolet/Chlorine Advanced Oxidation on the Production of Disinfection by-Products from Seven Model Benzene Precursors

**Wanting Li [1], Shihu Shu [1], Yanping Zhu [1,\*], Linjing Wu [1], Qiongfang Wang [2] and Naiyun Gao [3]**

[1] College of Environmental Science and Engineering, Donghua University, Shanghai 201620, China
[2] College of Chemistry and Chemical Engineering, Shanghai University of Engineering Science, Shanghai 201600, China
[3] State Key Laboratory of Pollution Control Reuse, Tongji University, Shanghai 200092, China
\* Correspondence: yanpingzhu@dhu.edu.cn

**Highlights:**

- MPUV/chlorine increased the chlorine demand and DBPFP of benzoic acid and nitrobenzene.
- MPUV/chlorine showed no further activation on active DBP precursors.
- MPUV/chlorine at pH 6 showed greater impact on precursors' DBPFP than at pH 8.
- MPUV/chlorine increased the BSF of THMs in the presence of bromine.

    Linear correlation was observed between changes in precursors' chlorine demand and DBPFP.

**Abstract:** UV/chlorine advanced oxidation process (AOP), as a potential alternative to UV/$H_2O_2$ in water treatment, may pose a potential risk of increased disinfection by-product (DBP) formation and is of great concern. In this paper, seven benzene derivatives, containing two chlorine-inert and five chlorine-active compounds, were selected as typical model DBP precursors, and the effects of medium pressure UV/chlorine (MPUV/chlorine) on their chlorine demand and DBP formation potential (DBPFP) were evaluated. The results showed that MPUV/chlorine could significantly increase the chlorine demand and DBPFP of the two inert precursors. For the four slow but active DBP precursors, MPUV/chlorine may accelerate their short-term DBP formation, whereas it showed an insignificant effect or even reduced their chlorine demand and DBPFP. For the only fast and active DBP precursor, MPUV/chlorine showed an insignificant effect on its short-term DBP formation or DBPFP. The overall effect of MPUV/chlorine was more significant at pH 6 than at pH 8. In the presence of $Br^-$, MPUV/chlorine significantly increased the bromine substitution factors of THMs. In addition, linear fitting results indicated that the UV/chlorine-induced change in overall chlorine demand may be considered as a potential indicator for the prediction of DBPFP alteration.

**Keywords:** medium pressure UV/chlorine; advanced oxidation process; disinfection by-products; benzene derivatives; bromide ($Br^-$)



## 1. Introduction

Ultraviolet (UV) irradiation and chlorination are commonly used disinfection strategies for the inactivation of water-borne pathogens. In recent years, with the rapid urbanization and industrialization of China, the problem of micro-pollution in source water has become prominent, and the removal of refractory micro-pollutants has posed a major challenge for conventional drinking water treatment processes. UV-based advanced oxidation process (AOP) can generate radicals via UV irradiation of oxidants and effectively remove refractory pollutants from water. Among the state-of-the-art UV-AOPs (e.g., $H_2O_2$, $O_3$, persulfate, chlorine, and peracetic acid) [1–6], UV/chlorine AOP can generate a diverse spectrum of reactive radicals including nonselective hydroxyl radicals (HO•) and selective

reactive chlorine species (RCS), such as Cl$^{\bullet}$, ClO$^{\bullet}$, and [7] Cl$_2{}^{\bullet-}$ [7,8], contributing to its comparable or even better performance in eliminating refractory micro-pollutants (e.g., pharmaceuticals and personal care products, endocrine disrupting chemicals, and algal toxins) compared to UV/H$_2$O$_2$ AOP [9–11]. Consequently, UV/chlorine AOP is emerging as a potential alternative to the conventional UV/H$_2$O$_2$ AOP for tertiary water treatment.

However, the relatively high chlorine dose used in UV/chlorine AOP compared to traditional disinfection (e.g., 2–10 mg/L vs. 0.2–2 mg/L) and the involvement of RCS oxidation may pose the risk of enhanced DBP formation during the process [12–14]. In addition, UV/chlorine oxidation may change the structure of organics in water while removing refractory pollutants, and thus affect the chlorine demand and disinfection by-product (DBP) formation of treated water during subsequent chlorine disinfection. Previous studies have found that the short-term (2–60 s) UV/chlorine treatment often does not directly generate a large amount of DBPs [15,16]. However, UV/chlorine oxidation may significantly increase the DBP formation potential (DBPFP) of effluent after the subsequent chlorine disinfection [17,18]. At present, relevant researches were mostly targeted to specific source water or natural organic matter (NOM) from specific sources [13,19–21]. Considering that the structure of DBP precursors in water is complex and varies greatly with different water sources, the research results have certain limitations. Therefore, this study attempts to disassemble a complex structure of aromatic precursors in the water body and select seven benzene derivatives with different substituents as the model precursors, in order to explore the relationship between the chemical structure of the precursors and the UV/chlorine-induced changing patterns of DBPs. Furthermore, in this study, medium pressure mercury lamp (MPUV) was used as UV light source, which has high power to obtain the UV dose for AOP within seconds and is widely applied in practical water plants, in order to better simulate the actual reaction conditions of UV/chlorine [13,22].

In summary, this study aimed to select seven benzene derivatives with different substituents as model DBP precursors, and evaluate the effect of MPUV/chlorine oxidation on their chlorine demand and DBP formation. Considering the pH-dependence of radical production in MPUV/chlorine system [23], both MPUV/chlorine at pH 6 and 8 were evaluated [24]. In addition, given that common co-existing bromide (Br$^-$) may promote the formation of highly toxic brominated disinfection by-products (Br$^-$DBPs), the effect of MPUV/chlorine on Br$^-$DBP formation in the presence of bromide was also investigated. The results of this study would provide insights into the relationship between MPUV/chlorine-induced DBP formation and precursor structure, and be helpful in predicting the DBP variation trend caused by MPUV/chlorine oxidation.

## 2. Materials and Methods

### 2.1. Chemicals and Materials

A total of seven benzene derivatives including phenol (PN), resorcinol (RSC), benzoic acid (BA), nitrobenzene (NB), o-chlorophenol (2-MCP), p-chlorophenol (4-MCP), and 2,4,6-trichlorophenol (2,4,6-TCP) were selected as model precursors, structures, and characteristics, which were shown in Table 1. Model precursors, sodium hypochlorite (NaOCl, 4.0–4.99%), and ascorbic acid were of analytical grade and obtained from Shanghai Titan Scientific Co., Ltd. (Shanghai, China). Analytical grade potassium hydrogen phosphate (K$_2$HPO$_4$), potassium dihydrogen phosphate (KH$_2$PO$_4$), sodium bromide, and anhydrous sodium sulfate (Na$_2$SO$_4$) were obtained from Sinopharm Chemical Reagent Co., Ltd. (Shanghai, China). All reaction solutions were prepared with Milli-Q$^{\circledR}$ ultrapure water. DBP standards, including THMs, HAAs, HANs, HALs, and TCNM were purchased from CanSyn Chemical Corp. (Toronto, ON, Canada). Internal standards (1,2-dibromopropane and 2,3,4,5-tetrafluorobenzoic acid) were purchased from Sigma-Aldrich (Saint Louis, MO, USA). HPLC grade methyl-tert-butyl ether (MTBE) and methanol were provided by Thermo Fisher (Waltham, MA, USA).

**Table 1.** List of seven benzene derivatives as model precursors.

| Name (Symbol) | Molecular Formula | Structure | Substituted Groups |
|---|---|---|---|
| Benzoic acid (BA) | $C_7H_6O_2$ | | -COOH |
| Nitrobenzene (NB) | $C_6H_5NO_2$ | | -NO$_2$ |
| Phenol (PN) | $C_6H_5OH$ | | -OH |
| Resorcinol (RSC) | $C_6H_6O_2$ | | -OH, -OH |
| o-chlorophenol (2-MCP) | $C_6H_5ClO$ | | -OH, -Cl |
| p-chlorophenol (4-MCP) | $C_6H_5ClO$ | | -OH, -Cl |
| 2,4,6-trichlorophenol (2,4,6-TCP) | $C_6H_3Cl_3O$ | | -OH, -Cl, -Cl, -Cl |

## 2.2. Experimental Procedures

### 2.2.1. Short-Term Oxidation in the UV Reactor

MPUV/chlorine illumination experiment was carried out by medium pressure UV quasi-parallel beam, as shown in Figure 1. Spectra of the MP mercury lamp (1 kW, Heraeus, Hanau, Germany) were shown in Figure S1. The average irradiance (10 mW/cm$^2$) was obtained using the calibrated Ocean Optics Spectro radiometer (USB4000), and the UV dose (600 mJ/cm$^2$, a typical dose used practically) [13,25,26] was obtained by multiplying the average irradiation intensity with the irradiation time. A 100 mL working solution containing a certain precursor (3.0 mg C/L) was dosed with 10 mg/L free chlorine and simultaneously irradiated at 22 ± 1 °C with stirring for 60 s. The reaction solution was adjusted to pH 6.0 or 8.0 using 10 mM phosphate buffer. Additionally, 1 mL of the solution was taken for the analysis of total chlorine residual after 60 s reaction. The effect of Br$^-$ on their DBP formation with regard to UV/chlorine treatment was evaluated by spiking 1 mg/L Br$^-$ to 100 mL precursor solution as discussed. Dark chlorination tests without UV irradiation were carried out as control under parallel conditions.

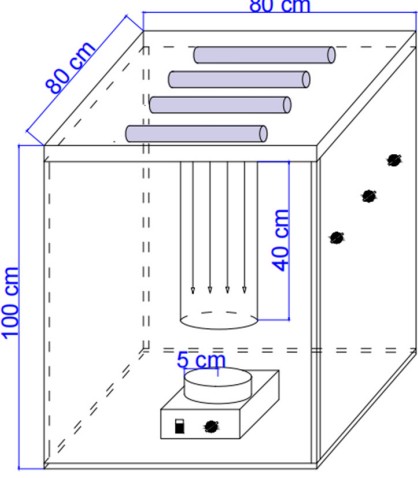

**Figure 1.** Schematic diagram of the ultraviolet collimated beam apparatus.

### 2.2.2. DBP Formation Potential Tests

After the pre-chlorination test, the samples were transferred to headspace-free amber glass bottles and subject to the recommended uniform formation condition (UFC) tests [27]. The working conditions were as follows: The pH value was adjusted to 8.0, and the initial residual chlorine value was adjusted by adding chlorine or quenching agent (ascorbic acid) to ensure that the residual chlorine was $1 \pm 0.4$ mg /L after standing at $22 \pm 1\ ^{\circ}$C for 24 h under dark conditions. The amount of chlorine or quench agent was determined by preliminary tests. Chlorine demand (total active chlorine) and DBP yields (i.e., DBPFP) were analyzed after 24 h post-chlorination. All experiments were conducted in duplicate with the averaged values being reported.

### 2.3. Analytical Methods

Concentrations of total chlorine in working solutions were determined using N,N diethyl-p-phenylenediamine (DPD) colorimetric method with a HACH DR/2500 spectrophotometer [28]. Additionally, 1,2-dibromopropane was used as the internal standard of THMs, HALs, HANs, and TCNM. Moreover, 2,3,4,5-tetrafluorobenzoic acid was used as the internal standard of HAA samples. After the addition of internal standards, samples of THMs, HALs, HANs, and TCNM were pretreated by liquid-liquid extraction according to USEPA 551.1 [29]. HAA samples were measured by GC-ECD after liquid-liquid extraction and derivatization according to USEPA 552.1 [30]. All DBPs were measured using a gas chromatograph (GC-2010, Shimadzu, Kyoto, Japan), which was equipped with HP-5 capillary column (30 m × 0.25 mm, 0.25 mm film thickness, J&W, USA) and electron capture detector. For HAAs, the temperature program was initially set at 40 $^{\circ}$C for 5 min, ramped to 65 $^{\circ}$C at 2.5 $^{\circ}$C/min, then to 85 $^{\circ}$C at 10 $^{\circ}$C/min and held for 3 min, and finally to 230 $^{\circ}$C at 20 $^{\circ}$C/min and held for 3 min. For non-HAA DBPs, the temperature program was set at 30 $^{\circ}$C for 10 min, ramped to 80 $^{\circ}$C at 7 $^{\circ}$C/min and held for 3 min, then to 110 $^{\circ}$C at 5 $^{\circ}$C /min and held for 3 min, and finally to 200 $^{\circ}$C at 20 $^{\circ}$C /min and held for 2.5 min. The limits of quantification (LOQ) for THMs, HAAs, HALs, HANs, and TCNM were less than 0.5, 1.5, 1.6, 2.4, and 0.05 μg/L. The effect of Br$^{-}$ on formation of brominated THMs and HAAs were analyzed by calculating the bromine substitution factors (BSFs) of trihalomethane and haloacetic acid (Equations (1) and (2)):

$$\text{BSF}_{\text{THMs}} = \frac{\sum_{n=1}^{3} n \times \text{CHCl}_{(3-n)}\text{Br}_n}{3 \times \sum_{n=0}^{n=3} \text{CHCl}_{(3-n)}\text{Br}_n} \tag{1}$$

$$\text{BSF}_{\text{HAAs}} = \frac{\sum_{n=1}^{3} \sum_{m=1}^{4-n} n \times \text{C}_2\text{H}_m\text{O}_2\text{Cl}_{(4-m-n)}\text{Br}_n}{\sum_{n=0}^{3} \sum_{m=1}^{3-n} (4-m) \times \text{C}_2\text{H}_m\text{O}_2\text{Cl}_{(4-m-n)}\text{Br}_n} \tag{2}$$

## 3. Results and Discussion

### 3.1. Chlorine Demand

The chlorine consumption of the seven model compounds in 60 s dark chlorination and MPUV/chlorine oxidation is shown in Table 2. During dark chlorination, RSC reacted with chlorine at an extremely fast rate among those precursors, with almost exhaustion of 10 mg/L chlorine within 60 s. However, the other six benzene derivatives reacted with chlorine relatively slowly, with the chlorine consumption ranging between 0.10 and 0.27 mg Cl/mg C. As UV was turned on, their chlorine consumption increased to different extents (except for RSC), which followed the order of NB < BA < 2,4,6-TCP < PN < 2-MCP < 4-MCP. The increase in chlorine consumption may be attributed to the photolysis of chlorine, and probably partly due to the higher reactivity of the intermediate products toward chlorine than the parent precursor in the MPUV/chlorine-treated system. The greater increase in chlorine consumption at pH 8 than at pH 6 may be related to the different dominant existing forms at different pHs (HOCl and OCl$^{-}$ at pH 6 and 8, respectively) and higher photolysis rate of OCl$^{-}$ than HOCl [31].

**Table 2.** Chlorine decay of the seven model compounds during the 60 s pre-oxidation under the following conditions: Chlorine dose = 10 mg $Cl_2$/L, pH = 6.0 or 8.0, [model benzene precursors]$_0$ = 3 mg-C/L, UV fluence rate = 10 mW/cm$^2$, [ $Br^-$ ] = 1 mg/L.

| Model Precursors | pH 6 (mg Cl/mg C) | | | | pH 8 (mg Cl/mg C) | | | |
|---|---|---|---|---|---|---|---|---|
| | Ambient | | Bromide-spiked | | Ambient | | Bromide-spiked | |
| | $Cl_2$ alone | UV/Chlorine | $Cl_2$ alone | UV/Chlorine | $Cl_2$ alone | UV/Chlorine | $Cl_2$ alone | UV/Chlorine |
| BA | 0.05 ± 0.0 | 0.37 ± 0.0 | 0.01 ± 0.0 | 0.38 ± 0.0 | 0.02 ± 0.0 | 0.96 ± 0.1 | 0.01 ± 0.0 | 0.98 ± 0.1 |
| NB | 0.01 ± 0.0 | 0.33 ± 0.0 | 0.01 ± 0.0 | 0.30 ± 0.0 | 0.01 ± 0.0 | 0.63 ± 0.0 | 0.01 ± 0.0 | 0.66 ± 0.0 |
| PN | 0.13 ± 0.0 | 0.93 ± 0.1 | 0.12 ± 0.0 | 0.98 ± 0.1 | 0.2 ± 0.0 | 1.34 ± 0.1 | 0.24 ± 0.0 | 1.47 ± 0.2 |
| RSC | 3.34 ± 0.3 | 3.38 ± 0.3 | 3.30 ± 0.3 | 3.37 ± 0.3 | 3.30 ± 0.2 | 3.37 ± 0.3 | 3.30 ± 0.3 | 3.37 ± 0.4 |
| 2-MCP | 0.23 ± 0.0 | 1.07 ± 0.0 | 0.53 ± 0.0 | 1.37 ± 0.0 | 0.27 ± 0.0 | 1.53 ± 0.1 | 0.55 ± 0.0 | 1.65 ± 0.1 |
| 4-MCP | 0.10 ± 0.0 | 1.33 ± 0.1 | 0.08 ± 0.0 | 1.57 ± 0.1 | 0.13 ± 0.0 | 1.73 ± 0.1 | 0.17 ± 0.0 | 1.83 ± 0.1 |
| 2,4,6-TCP | 0.20 ± 0.0 | 0.70 ± 0.0 | 0.23 ± 0.0 | 1.37 ± 0.1 | 0.17 ± 0.0 | 1.15 ± 0.1 | 0.21 ± 0.0 | 1.37 ± 0.1 |

The overall chlorine demands of the seven model compounds after post-chlorination were shown in Table 3. Compared with dark chlorination, MPUV/chlorine oxidation showed an activation effect on the two inert precursors (i.e., BA and NB) especially at pH 6, which increased the chlorine demands of BA and nitrobenzene from 0.02–0.05 mg Cl/mg C and 0.01 mg Cl/mg C to 1.50–2.07 mg Cl/mg C and 0.97–1.23 mg Cl/mg C, respectively. The greater activation effect at pH 6 indicated the probably higher contribution of HO$^\bullet$/Cl$^\bullet$ than ClO$^\bullet$ considering that the concentration of HO$^\bullet$/Cl$^\bullet$ in the system was higher under acidic conditions than under basic conditions [32]. On the contrary, MPUV/chlorine oxidation reduced or showed little effect on the chlorine demands of the five chlorine-active precursors. Additionally, MPUV/chlorine oxidation promoted the chlorination rate of these five precursors (Table 2), but was not likely to further increase their overall chlorine demands. Moreover, it can be seen from Tables 2 and 3 that addition of 1 mg/L Br$^-$ may increase the chlorine demands of certain precursors (e.g., PN and 2-MCP), which may be probably due to the consumption of chlorine via the oxidation of Br$^-$ by chlorine [33]. However, Br$^-$ addition showed a minor effect on the overall changing trends of these precursors' short-term chlorine decay or chlorine demand.

**Table 3.** Chlorine demands of the seven model compounds after the 60 s short-term oxidation under the following conditions: Chlorine dose = 10 mg $Cl_2$/L, pH = 6.0 or 8.0, [model benzene precursors]$_0$ = 3 mg-C/L, UV fluence rate = 10 mW/cm$^2$, [Br$^-$] = 1 mg/L.

| Model Precursors | pH 6 (mg Cl/mg C) | | | | pH 8 (mg Cl/mg C) | | | |
|---|---|---|---|---|---|---|---|---|
| | Ambient | | Bromide-spiked | | Ambient | | Bromide-spiked | |
| | $Cl_2$ alone | UV/Chlorine | $Cl_2$ alone | UV/Chlorine | $Cl_2$ alone | UV/Chlorine | $Cl_2$ alone | UV/Chlorine |
| BA | 0.05 ± 0.0 | 2.07 ± 0.1 | 0.12 ± 0.0 | 2.07 ± 0.1 | 0.02 ± 0.0 | 1.50 ± 0.1 | 0.10 ± 0.0 | 1.50 ± 0.1 |
| NB | 0.01 ± 0.0 | 0.97 ± 0.1 | 0.01 ± 0.0 | 1.01 ± 0.0 | 0.01 ± 0.0 | 1.23 ± 0.1 | 0.01 ± 0.0 | 1.29 ± 0.1 |
| PN | 6.71 ± 0.3 | 6.47 ± 0.2 | 7.42 ± 0.2 | 6.89 ± 0.2 | 6.90 ± 0.2 | 6.66 ± 0.2 | 7.68 ± 0.1 | 6.78 ± 0.2 |
| RSC | 6.11 ± 0.2 | 6.32 ± 0.3 | 6.27 ± 0.2 | 6.31 ± 0.3 | 6.30 ± 0.1 | 6.45 ± 0.0 | 6.42 ± 0.0 | 6.54 ± 0.1 |
| 2-MCP | 6.87 ± 0.3 | 6.72 ± 0.1 | 7.40 ± 0.2 | 7.22 ± 0.1 | 7.59 ± 0.2 | 6.81 ± 0.2 | 7.93 ± 0.2 | 7.36 ± 0.1 |
| 4-MCP | 7.19 ± 0.2 | 6.65 ± 0.2 | 7.14 ± 0.2 | 7.2 ± 0.1 | 7.29 ± 0.3 | 6.76 ± 0.0 | 7.63 ± 0.1 | 7.14 ± 0.2 |
| 2,4,6-TCP | 6.72 ± 0.2 | 6.17 ± 0.1 | 6.71 ± 0.1 | 6.22 ± 0.2 | 6.10 ± 0.0 | 6.30 ± 0.1 | 6.43 ± 0.1 | 6.25 ± 0.0 |

In summary, for benzene precursors with low chlorination rate but high chlorine demand, MPUV/chlorine tends to promote their chlorine consumption rate but reduce their chlorine demand, while for chlorine-inert precursors, MPUV/chlorine showed a significant activation effect, increasing both their chlorination rate and overall chlorine demands especially under acidic conditions.

### 3.2. Effect of MPUV/Chlorine Oxidation on DBP Formation from Seven Benzene Precursors

3.2.1. Effect of MPUV/Chlorine Oxidation on DBP Formation from Inert Benzene Precursors without Bromide

As shown in Figure 2, the 60 s short-term DBP formation and DBPFP of BA and NB were within 5 µg/mg C, indicating that both BA and NB are typical inert DBP precursors. Turning on UV increased their short-term DBP formation to 8.41–8.41 µg/mg C and 5.91–7.62 µg/mg C, respectively, and their DBPFP to 20.98–42.47 µg/mg C and 17.45–33.53 µg/mg C, with more significant activation due to MPUV/chlorine oxidation at pH 6 than at pH 8. The different activation extents of the two precursors may be attributed to their different reactivity with HO$^\bullet$ and RCS$^\bullet$, for example, BA has a high reactivity with both HO$^\bullet$ and Cl$^\bullet$, while NB mainly reacts with HO$^\bullet$ [7,34]. In terms of specific DBP species, after 60 s MPUV/chlorine and post-chlorination, BA mainly formed 18.71–18.72 µg THMs/mg C, 13.73–27.47 µg HAAs/mg C, and 0.53–1.28 µg HALs/mg C, while NB produced 4.00–5.22 µg THMs/mg C, 1.74–4.52 µg HALs/mg C, and 7.41 µg TCNM/mg C. Prior study found that HO$^\bullet$ can convert BA to salicylic acid, which is more chlorine-reactive and was considered as an important HAA and/or HAL precursor [35,36]. Nitrobenzene would be converted to nitrophenol during the reaction with HO$^\bullet$, and nitrophenol further generated DBPs, such as THMs, HALs, and TCNM by chlorination [35]. Among them, the significant promotion of TCNM formation due to MPUV/chlorine oxidation of NB in this study is consistent with the previously reported finding that UV/chlorine treatment may potentially lead to increased TCNM formation of nitrogenous precursors [37], which needs to be of concern.

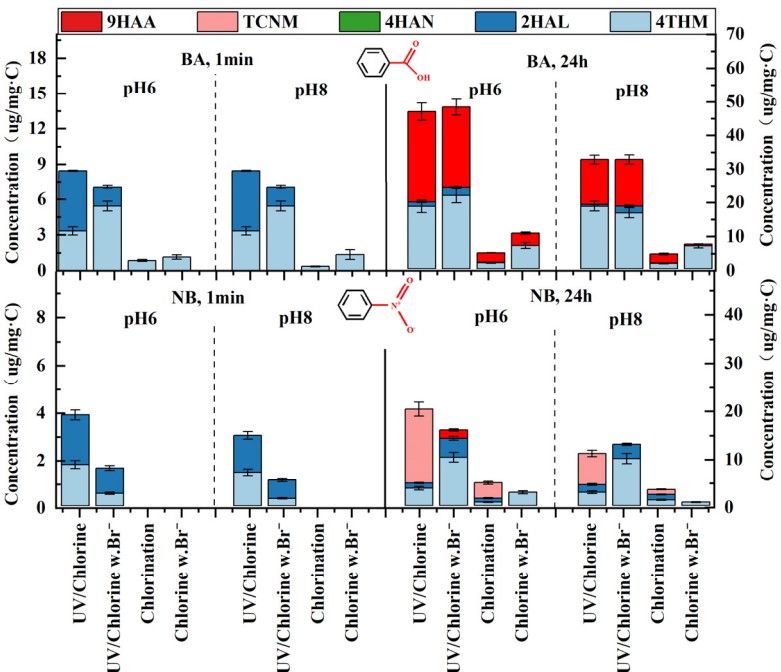

**Figure 2.** The formation of 4THM, 9HAA, 4HAN, 2HAL, and TCNM during the 1 min short-term oxidation and 24 h post-chlorination from two inert benzene derivatives. Short-term oxidation conditions: Chlorine dose = 10 mg chlorine/L, pH = 6.0 or 8.0, [model benzene precursors]$_0$ = 3 mg-C/L, UV fluence rate = 10 mW/cm$^2$, [bromide]$_0$= 1 mg/L. Post-chlorination conditions: pH = 8.0, chlorination time = 24 h, chlorine residual = 1.0 ± 0.4 mg/L, dark, 22 °C.

3.2.2. Effects of Bromide on DBP Formation from Inert Benzene Precursors

The coexisting Br$^-$ in water would react rapidly with HOCl to form HOBr (Reaction 1) [38] and potentially further generate Br$^-$DBPs due to the bromination reaction between HOBr and organic matter [39]. During MPUV/chlorine oxidation, HOBr

photolysis would generate reactive bromine radicals (RBS) and HO$^\bullet$ (Reaction 2) [40], and on the other side would consume HO$^\bullet$ via Reaction (3) [33]. Nevertheless, the coexistence of bromide may affect the composition of radical species during the MPUV/chlorine treatment. The effect of bromide on the DBP formation of BA and NB in the MPUV/chlorine-treated system was shown in Figure 3. For BA, the coexisting bromide slightly promoted the production of DBAA and BCAA during dark chlorination and post-chlorination. With UV on, the coexisting bromide significantly increased the production of Br$^-$THMs (DBCM and TBM) and Br$^-$HAAs (DBAA and BCAA) by 10.34–14.42 µg/mg C and 4.30–5.33 µg/mg C, respectively, although this caused a minor change in the total DBPFP. Similarly, the coexisting bromide caused rare Br$^-$THM formation of NB in the dark, while, with UV on, this promoted the formation of 3.40–3.58 µg DBCM/mg C and 6.83–8.71 µg TBM/mg C, respectively. This promotion effect was probably attributed to the reaction between RBS/HOBr and precursors/intermediates [33].

$$HClO + Br^- \rightarrow HOBr + Cl^- \tag{R1}$$

$$HOBr \xrightarrow{uv} HO^\bullet + Br^\bullet \tag{R2}$$

$$HOBr + HO^\bullet \rightarrow BrO^\bullet + H_2O \tag{R3}$$

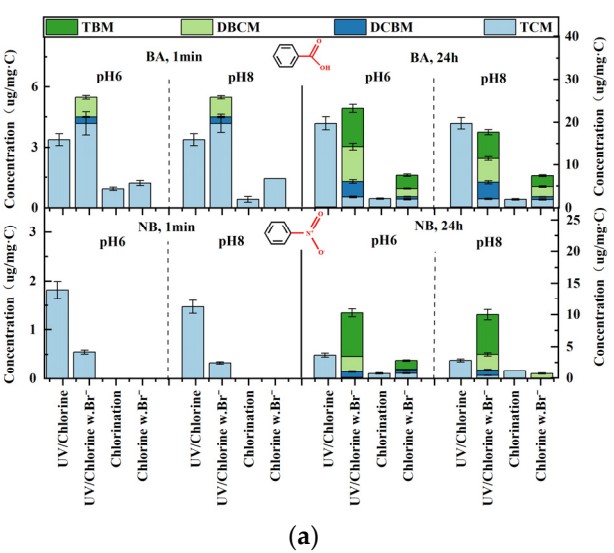
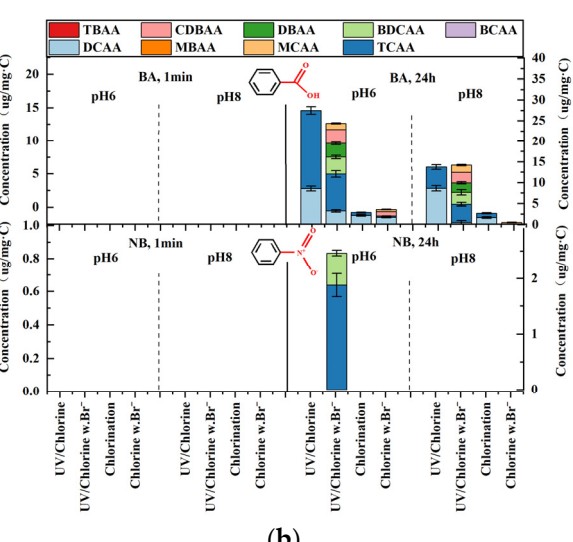

(a)  (b)

**Figure 3.** The formation of THMs (**a**) and HAAs (**b**) during the 1 min short-term oxidation and after 24 h post-chlorination from two inert benzene derivatives. Short-term oxidation conditions: Chlorine dose = 10 mg chlorine /L, pH = 6.0 or 8.0, [model benzene precursors]$_0$ = 3 mg-C/L, UV fluence rate = 10 mW/cm$^2$, [bromide]$_0$= 1 mg/L. Post-chlorination conditions: pH = 8.0, chlorination time = 24 h, chlorine residual = 1.0 ± 0.4 mg/L, dark, 22 °C.

The BSFs of THMFP and HAAFP under different treatment conditions were shown in Table 4. Turning on UV increased BSF$_{THMs}$ of BA and NB from 1.79–1.85 and 1.76–2.00 to 1.93–2.03 and 2.45–2.55, respectively, and BSF$_{HAAs}$ from 0.73–1.00 and 0.00 to 0.81–1.04 and 0.00–0.48, respectively, compared with dark chlorination. This trend was more significant under pH 6 than pH 8. Overall, the effect of coexisting bromide on BSF$_{THMs}$ was greater relative to BSF$_{HAAs}$, which is consistent with the reported bromide effect on actual water and model amino acid precursors in prior UV/chlorine studies [17,41].

Table 4. Bromine incorporation factors of THMs and HAAs from the seven benzenes under the following conditions: Chlorine dose = 10 mg $Cl_2$/L, pH = 6.0 or 8.0, [model benzene precursors]$_0$ = 3 mg-C/L, UV fluence rate = 10 mW/cm$^2$, [$Br^-$] = 1 mg/L.

| Benzene Derivatives | Cl/Br(mg/mg-C) | | | | UV/Cl/Br(mg/mg-C) | | | |
|---|---|---|---|---|---|---|---|---|
| | PH6 | | PH8 | | PH6 | | PH8 | |
| | THM | HAA | THM | HAA | THM | HAA | THM | HAA |
| BA | 1.85 | 0.73 | 1.78 | 1.00 | 2.03 | 0.81 | 1.93 | 1.04 |
| NB | 1.76 | 0.00 | 2.00 | 0.00 | 2.55 | 0.48 | 2.45 | 0.00 |
| PN | 1.13 | 0.74 | 1.19 | 0.77 | 1.33 | 0.75 | 1.26 | 0.78 |
| RSC | 0.28 | 1.08 | 0.33 | 1.04 | 0.56 | 1.10 | 0.47 | 1.06 |
| 2-MCP | 1.25 | 0.38 | 1.33 | 0.34 | 1.47 | 0.44 | 1.51 | 0.47 |
| 4-MCP | 1.30 | 0.26 | 1.20 | 0.27 | 1.44 | 0.28 | 1.33 | 0.28 |
| 2,4,6-TCP | 1.39 | 0.29 | 1.21 | 0.28 | 1.52 | 0.35 | 1.23 | 0.35 |

### 3.3. Effect of MPUV/Chlorine Oxidation on DBP Formation from Reactive Benzene Precursors

3.3.1. Effect of MPUV/Chlorine Oxidation on DBP Formation from Reactive Benzene Precursors without Bromide

The DBP formation during chlorination of the contained phenolic compounds first reacted with benzene ring to form chlorine substitution products, and then disinfection by-products, such as trihalomethanes and haloacetic acids through ring opening and hydrolysis [42]. The DBP formation and DBPFP of five highly chlorine-reactive benzene derivatives were shown in Figure 4. As a significant DBP precursor with high chlorine-reactivity, RSC formed 29.81–73.14 μg THMs/mg C and 1.68–1.96 μg HAAs/mg C during 60 s dark chlorination, and 356.82–422.99 μg THMFP/mg C, 29.71–47.41 μg HAAFP/mg C, and 3.05–4.12 μg HALFP/mg C after post-chlorination. The values were comparable to the reported DBPFP of RSC [43]. Turning on UV showed no significant effect on its 60 s short-term DBP formation or its DBPFP. This is consistent with UV/chlorine-induced changing trends of its chlorine demand (Table 4), indicating that UV/chlorine would not likely accelerate or increase DBP formation of precursors with extremely high chlorine reactivity and fast chlorination rate. Similar DBP changing trends were also observed in our previous work on active amino acid-type precursors [41]. For DBP species, HAAFP increased by 62.84–91.77 μg/mg C probably at the expense of decreased THMFP (by 89.56–106.43 μg/mg C).

The four slow DBP precursors showed very limited DBP formation (0–44.88 μg/mg C) during the 60 s dark chlorination but high DBPFP (474.95–1071.96 μg/mg C) after post-chlorination. Their DBPFP increased with the number of chlorine substitutions on the benzene ring; 2,4,6-TCP had the highest DBPFP, followed by 2-MCP and 4-MCP, and PN the lowest. Turning on UV accelerated their DBP formation within 60 s, while it showed a minor effect or even slight deactivation on their DBPFP. For example, the 60 s DBP formation of 2,4,6-TCP increased from 44.88 to 37.07 μg/mg C, while its DBPFP slightly decreased from 1071.96 to 862.11 μg/mg C, with a more significant effect at pH 6.0 than at pH 8.0. For the variation of individual DBP species, short-term THM and HAA formation in the reactor increased from 0 to 12.01–15.82 μg/mg C and 24.28–28.33 μg/mg C, respectively, due to UV/chlorine oxidation, whereas the HAAFP and HALFP increased from 377.72 and 2.03 μg/mg C to 899.08 and 37.07 μg/mg C, respectively, along with THMFP decreasing from 270.01 to 41.17 μg/mg C. The enhanced HAA and/or HAL formation has been previously observed in HO$^\bullet$-based AOP systems [44,45]. Generally, the effect of UV/chlorine under acidic pH conditions was stronger than under basic conditions, which indirectly reflected the greater contribution of HO$^\bullet$ and/or Cl$^\bullet$ than ClO$^\bullet$ to the DBP changes. Previous studies on the mechanism of radical action also found that the reaction mechanism of HO$^\bullet$ and Cl$^\bullet$ which were dominant radicals under acidic conditions is mainly an addition and electron transfer, while the ClO$^\bullet$ with relatively high concen-

trations under alkaline conditions is mainly an electron transfer, and thus may lead to a relatively smaller effect on DBP production under alkaline conditions [14,46–48].

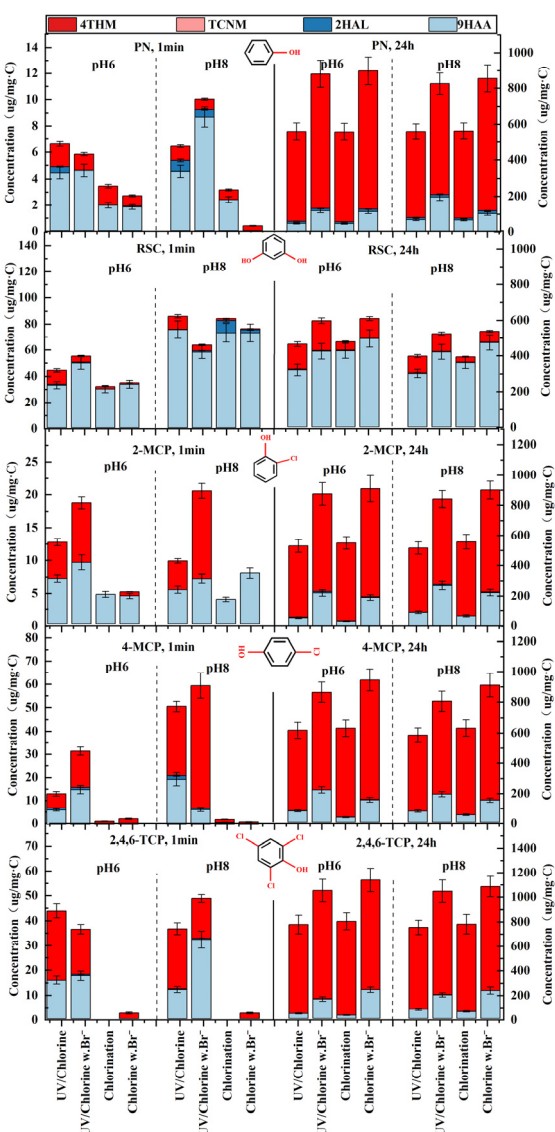

**Figure 4.** The formation of 4THM, 9HAA, 2HAL, and TCNM during the 1 min short-term oxidation and 24 h post-chlorination from five active benzene derivatives. Short-term oxidation conditions: Chlorine dose = 10 mg chlorine/L, pH = 6.0 or 8.0, [model benzene precursors]$_0$ = 3 mg-C/L, UV fluence rate = 10 mW/cm$^2$, [bromide]$_0$= 1 mg/L. Post-chlorination conditions: pH = 8.0, chlorination time = 24 h, chlorine residual = 1.0 ± 0.4 mg/L, dark, 22 °C.

Above all, MPUV/chlorine-induced DBP changes in the five active precursors were generally consistent with the corresponding changes in their chlorine consumption. The results indicated that MPUV/chlorine oxidation would not likely contribute to higher DBPFP or chlorine demand for active precursors, although it may accelerate the DBP formation rate of these slow DBP precursors.

### 3.3.2. Effect of Bromide on DBP Formation from Reactive Benzene Precursors

The effect of bromide on the THM and HAA formation of the five active benzenes was shown in Figures 5 and 6. Generally, the presence of bromide increased the overall DBPFP of the five precursors due to the formation of bromide-contained DBPs during dark chlorination. Turning on UV showed an insignificant impact on the overall DBP

changing trend in bromide-spiked systems although this caused varied DBP changing extents relative to non-bromide systems. Specifically, for PN and RSC, in the presence of bromide, minor brominated THMs/HAAs were formed within 60 s with UV on or off, while for the three Cl-substituted phenol precursors, bromide coexistence promoted brominated THM and HAA formation especially with UV on, indicating the possible contribution of RBS to the halogenation of Cl-substituted phenol precursors. Nevertheless, UV/chlorine may accelerate the brominated DBP formation for certain chlorine-substituted benzene precursors. However, after post-chlorination, the presence of bromide consistently increased their THMFP and HAAFP under dark conditions, and turning on UV did not further promote brominated DBP formation compared with the dark control. For example, for 2,4,6-TCP, turning on UV increased the production of THMs and HAAs in 60 s in the presence of bromide from 0.00 µg/mg C and 2.30–2.30 µg/mg C to 17.55–32.35 µg/mg C and 16.26–18.53 µg/mg C, but slightly decreased its THMFP and HAAFP by 175.68–212.62 µg/mg C and 125.44–134.39 µg/mg C, respectively.

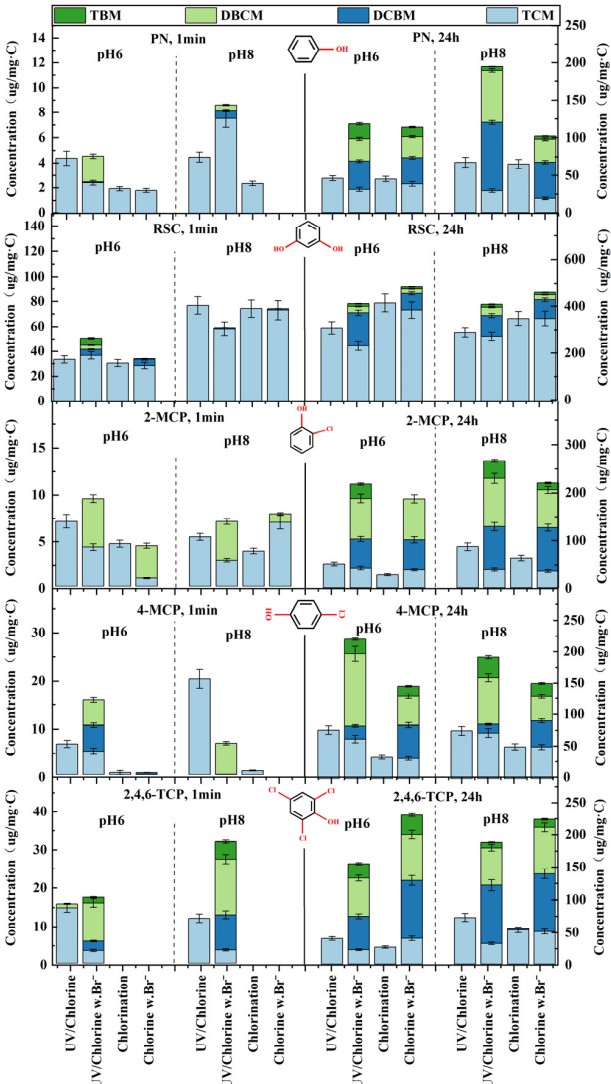

**Figure 5.** The formation of 4THM during the 1 min short-term oxidation and 24 h post-chlorination from five active benzene derivatives. Short-term oxidation conditions: Chlorine dose = 10 mg chlorine/L, pH = 6.0 or 8.0, [model benzene precursors]$_0$ = 3 mg-C/L, UV fluence rate = 10 mW/cm$^2$, [bromide]$_0$ = 1 mg/L. Post-chlorination conditions: pH = 8.0, chlorination time = 24 h, chlorine residual = 1.0 ± 0.4 mg/L, dark, 22 °C.

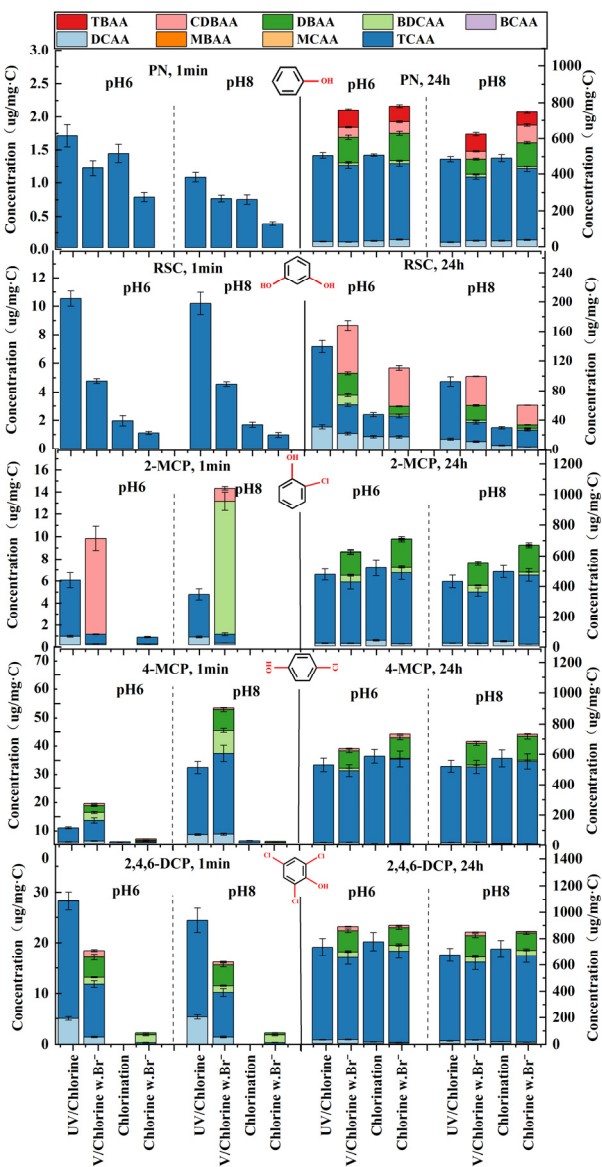

**Figure 6.** The formation of 9HAA during the 1 min short-term oxidation and 24 h post-chlorination from five active benzene derivatives. Short-term oxidation conditions: Chlorine dose = 10 mg chlorine/L, pH = 6.0 or 8.0, [model benzene precursors]$_0$ = 3 mg-C/L, UV fluence rate = 10 mW/cm$^2$, [bromide]$_0$ = 1 mg/L. Post-chlorination conditions: pH = 8.0, chlorination time = 24 h, chlorine residual = 1.0 ± 0.4 mg/L, dark, 22 °C.

The BSF$_{THMs}$ and BSF$_{HAAs}$ of the five precursors under different treatment conditions were shown in Table 4. UV/chlorine oxidation increased the BSF$_{THMs}$ of PN, RSC, 2-MCP, 4-MCP, and 2,4,6-TCP from 1.13–1.19, 0.28–0.33, 1.25–1.33, 1.20–1.30, and 1.21–1.39, to 1.26–1.33, 0.47- 0.56, 1.47–1.51, 1.33–1.44, and 1.23–1.52, respectively. In contrast, BSF$_{HAAs}$ remained almost unchanged. This was basically consistent with the BSF changing trends of the two inert benzene precursors in Section 3.2.2, further reflecting the effect of MPUV/chlorine on brominated THM formation in bromide-contained systems.

### 3.4. Relationship between MPUV/Chlorine-Induced Chlorine Demand Changes and Corresponding DBPFP Changing Trends of Model Precursors

To help in the prediction of MPUV/chlorine-induced changing trends in treated water's DBPFP (Δ(DBPFP)), the linear correlation between the change in chlorine demand (Δ(chlorine demand)) of the seven model precursors and their DBPFP changes (Δ(DBPFP))

was fitted and shown in Figure 7. A positive correlation was observed in the seven precursors' Δ(chlorine demand) and their Δ(DBPFP) under each treatment condition, with $R^2$ values distributed between 0.70 and 0.86. Therefore, considering that the measurement of chlorine demand is more convenient and cost-effective than DBPFP, the change in chlorine demand may be considered as a potential indicator to predict the effect of MPUV/chlorine AOP treatment on treated water's DBPFP and help in the determination of whether the process may pose a potential threat to DBPFP.

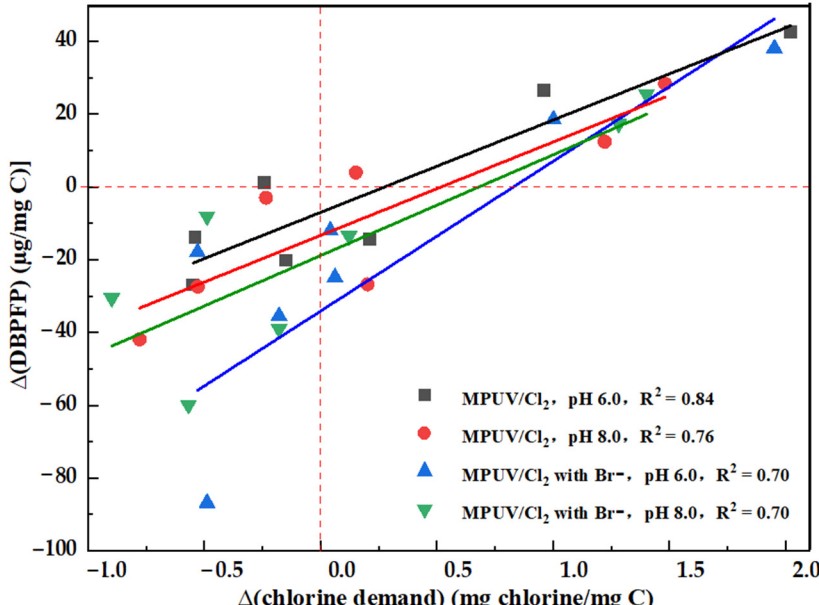

**Figure 7.** Linear fitting between Δ(chlorine demand) and Δ(DBPFP) caused by MPUV/chlorine AOP oxidation.

## 4. Conclusions

In this paper, the relationship between the DBP changing pattern and the structural characteristics of precursors was investigated by evaluating MPUV/chlorine oxidation-induced alteration of DBP formation and DBPFP from seven model benzene precursors containing different substituents. According to the results in this study, for the two inert precursors with low chlorine demand, MPUV/chlorine activated them and increased their chlorine demand and DBPFP, with the activation more significant under acidic conditions. In contrast, for active DBP precursors, MPUV/chlorine showed no significant effect or even exhibited slight deactivation on their DBPFP. For the benzene precursors with low chlorination rate but high chlorine demand, MPUV/chlorine accelerated their chlorine consumption and DBP formation rates, but did not further increase their chlorine demand or DBPFP. For bromide-spiked samples, the results showed that bromide addition consistently increased their THMFP and HAAFP under dark conditions, and turning on UV did not further promote brominated DBP formation compared with the dark control. Bromide caused significant increases in $BSF_{THMs}$ of the seven substances, which should be of concern. MPUV/chlorine-induced changes in the seven precursors' chlorine demand was linearly correlated to their DBPFP changes, indicating that Δ(chlorine demand) may be considered as a potential indicator in the prediction of the changing trends of DBPFP for MPUV/chlorine-treated water.

**Supplementary Materials:** The following supporting information can be downloaded at: https://www.mdpi.com/article/10.3390/w14223775/s1, Figure S1: spectrogram of a medium-pressure mercury lamp; Table S1: Chlorine decay during 60 s pre-oxidation of the seven model compounds.

**Author Contributions:** W.L.: Data curation, Writing—Original draft preparation; S.S.: Visualization, Investigation, Funding acquisition; Y.Z.: Conceptualization, Writing—Original draft preparation, Supervision, Funding acquisition; L.W.: Data curation, Writing—Original draft preparation; Q.W.: Methodology, Writing—Reviewing and Editing; N.G.: Writing—Reviewing and Editing. All authors have read and agreed to the published version of the manuscript.

**Funding:** This research received no external funding.

**Acknowledgments:** This work was supported by the National Natural Science Foundation of China (No. 52000023), Shanghai Committee of Science and Technology (No. 19DZ1204400), Fundamental Research Funds for Center Universities (21D111311), and the Shanghai Sailing Program (20YF1401200).

**Conflicts of Interest:** The authors declare no conflict of interest.

## Abbreviations

DBPs, disinfection by-products; DBPFP, DBP formation potential; THMs, trihalomethanes; TCM, trichloromethane; BDCM, bromodichloromethane; DBCM, dibromochloromethane; TBM, tribromomethane; HAAS, haloacetic acids; MCAA, monochloroacetic acid; MBAA, monobromoacetic acid; DCAA, dichloroacetic acid; DBAA, dibromoacetic acid; BCAA, bromochloroacetic acid; TCAA, trichloroacetic acid; BDCAA, bromodichloroacetic acid; CDBAA, chlorodibromoacetic acid; TBAA, tribromoacetic acid; HANs, haloacetonitriles; DCAN, dichloroacetonitrile; BCAN, bromochloroacetonitrile; DBAN, dibromoacetonitrile; TCAN, trichloroacetonitrile; HALS, haloacetaldehydes; DCAL, dichloroacetaldehyde; CH, trichloroacetaldehyde; TCNM, trichloronitromethane; RCS, reactive chlorine species; NOM, natural organic matter; PN, phenol; RSC, resorcinol; BA, benzoic acid; NB, nitrobenzene; 2-MCP, o-chlorophenol; 4-MCP, p-chlorophenol; 2,4,6-TCP, 2,4,6-trichlorophenol; MTBE, methyl-tert-butyl ether; UFC, uniform formation condition; BSFs, bromine substitution factors.

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
