# Peer review of "Effect of Medium Pressure Ultraviolet/Chlorine Advanced Oxidation on the Production of Disinfection by-Products from Seven Model Benzene Precursors"

_water, doi:10.3390/w14223775_

Round 1

Reviewer 1 Report

The authors present their work on the Effect of medium-pressure ultraviolet/chlorine advanced oxidation on the production of disinfection by-products from seven 3-model benzene precursors. Here are my few suggestions:

1. There is a lot of literature on UV-based oxidation technologies. The introduction does not provide a sufficient review of the technology and the advances made in this field. 

2. Analytical Methods need QA/QC and more details on the methods.

3. ph 6 and 8 are very close, Table 2 doesnt provide full details, caption need to include what those numbers are etc.

4. What was the basis for selecting 600mJ/cm^2 UV dosages? Was there any optimization performed? This needs to be either explained or resolved.

5. Results section needs a significant improvement in terms of comparison to other studies. currently, there are very few in this section.

Author Response

The authors present their work on the Effect of medium-pressure ultraviolet/chlorine advanced oxidation on the production of disinfection by-products from seven 3-model benzene precursors. Here are my few suggestions:

  1. There is a lot of literature on UV-based oxidation technologies. The introduction does not provide a sufficient review of the technology and the advances made in this field.

Response: Thanks for the suggestion. As requested, we have added several sentences in lines 34-37 to introduce the UV-based advanced oxidation process briefly. Also, several related references have been added to lines 37 and 42. The added sentences in lines 34-37 are as follows:

UV-based advanced oxidation process (AOP) can generate radicals via UV irradiation of oxidants and effectively remove refractory pollutants from water . Among those state of the art UV-AOPs (e.g., H2O2, O3, persulfate, chlorine, and peracetic acid),

The added references in line 37 and 42 are as follows:

[1] D.B. Miklos, W.L. Wang, K.G. Linden, J.E. Drewes, U. Hübner, Comparison of UV-AOPs (UV/H2O2, UV/PDS and UV/Chlorine) for TOrC removal from municipal wastewater effluent and optical surrogate model evaluation[J]. Chemical Engineering Journal, 2019, 362, 537-547.

[2] Y.L. Zhang, W.L. Wang, M.Y. Lee, Z.W. Yang, Q.Y. Wu, N. Huang, H.Y. Hu, Promotive effects of vacuum-UV/UV (185/254 nm) light on elimination of recalcitrant trace organic contaminants by UV-AOPs during wastewater treatment and reclamation: A review[J]. Science of The Total Environment, 2022, 818, 151776.

[3] L. Varanasi, E. Coscarelli, M. Khaksari, L.R. Mazzoleni, D. Minakata, Transformations of dissolved organic matter induced by UV photolysis, Hydroxyl radicals, chlorine radicals, and sulfate radicals in aqueous-phase UV-Based advanced oxidation processes[J]. Water Research, 2018, 135, 22-30.

[4] J. Wang, J. Deng, E. Du, H. Guo, Reevaluation of radical-induced differentiation in UV-based advanced oxidation processes (UV/hydrogen peroxide, UV/peroxydisulfate, and UV/chlorine) for metronidazole removal: Kinetics, mechanism, toxicity variation, and DFT studies[J]. Separation and Purification Technology, 2022, 301, 121905.

[5] J.D. Laat, M.I. Stefan, UV/chlorine process in M.I. Stefan, Ed., Advanced oxidation processes for water treatment, IWA Publishing, London, UK, Chapter 9, 2018.

[6] K. Guo, Z. Wu, C. Chen, J. Fang, UV/Chlorine process: An efficient advanced oxidation process with multiple radicals and functions in water treatment[J]. Accounts of Chemical Research, 2022, (3): 55.

[9] Y.Q. Gao, J. Zhang, C. Li, F.X. Tian, N.Y. Gao, Comparative evaluation of metoprolol degradation by UV/chlorine and UV/H2O2 processes[J]. Chemosphere, 2020, 243, 125325.

[10] K. Guo, Z. Wu, S. Yan, B. Yao, W. Song, Z. Hua, X. Zhang, X. Kong, X. Li, J. Fang, Comparison of the UV/chlorine and UV/H2O2 processes in the degradation of PPCPs in simulated drinking water and wastewater: Kinetics, radical mechanism and energy requirements[J]. Water Research, 2018, 147, 184-194.

[11] C.W. Pai, G.S. Wang, Treatment of PPCPs and disinfection by-product formation in drinking water through advanced oxidation processes: Comparison of UV, UV/Chlorine, and UV/H2O2[J]. Chemosphere, 2022, 287, 132171.

  1. Analytical Methods need QA/QC and more details on the methods.

Response: The QA/QC had been done in our study and more description on the quantification of DBPs were added in section 2.3. The added sentences in lines 155-165 were as follows:

All DBPs were measured using a gas chromatograph (GC-2010, Shimadzu, Japan ), which was equipped with a HP-5 capillary column (30 m x 0.25 mm., 0.25 mm film thickness, J&W, USA) and an electron capture detector. For HAAs, the temperature program was initially set at 40 oC for 5 min, ramped to 65 oC at 2.5 oC/min, then to 85 oC at 10℃/min and held for 3 min, and finally to 230 oC at 20 oC /min and held for 3 min. For non-HAA DBPs, the temperature program was set at 30 oC for 10 min, ramped to 80 oC at 7 oC /min and held for 3 min, then to 110 oC at 5 oC /min and held for 3 min, and finally to 200 oC at 20 oC /min and held for 2.5 min. The limits of quantification (LOQ) for THMs, HAAs, HALs, HANs, and TCNM were less than 0.5, 1.5, 1.6, 2.4 and 0.05 µg/L.

  1. ph 6 and 8 are very close, Table 2 doesn’t provide full details, caption need to include what those numbers are etc.

Response: Generally, the relevant pH of drinking or reclaimed water was between 6 and 8[24], and considering the pKa value of HOCl/OCl- was 7.5, pH 6 and 8 were chosen in our experiment.

As requested, we have added specific treatment conditions to its caption as follows:

Table 3. Chlorine demands of the seven model compounds after 60 s short-term oxidation under conditions: chlorine dose = 10 mg Cl2/L, pH = 6.0 or 8.0, [model benzene precursors]0 = 3 mg-C/L, UV fluence rate = 10 mW/cm2, [Br-] = 1 mg/L.

[24] C. Wang, Z. Ying, M. Ma, M. Huo, W. Yang, Degradation of micropollutants by UV chlorine treatment in reclaimed water: pH effects, formation of disinfectant byproducts, and toxicity assay, Water, 2019.

  1. What was the basis for selecting 600 mJ/cm2 UV dosages? Was there any optimization performed? This needs to be either explained or resolved.
    Response: As reported, 600 mJ/cm2 is the typical UV dosage used for practical UV-AOP treatment of drinking/reclaimed water [13,26]. The relevant content and reference have been added to line 98.

[13] D. Wang, J.R. Bolton, S.A. Andrews, R. Hofmann, Formation of disinfection by-products in the ultraviolet/chlorine advanced oxidation process[J]. Science of The Total Environment, 2015, 518-519, 49-57.

[26] Y.H. Chuang, A. Szczuka, F. Shabani, J. Munoz, R. Aflaki, S.D. Hammond, W.A. Mitch, Pilot-scale comparison of microfiltration/reverse osmosis and ozone/biological activated carbon with UV/hydrogen peroxide or UV/free chlorine AOP treatment for controlling disinfection byproducts during wastewater reuse[J]. Water Research, 2019, 152, 215-225.

  1. Results section needs a significant improvement in terms of comparison to other studies. currently, there are very few in this section.
    Response: Thanks for your suggestion. As requested, two more comparisons have been added to the results section. Specifically, in lines 273-274, the sentence “Similar DBP changing trends were also observed in our previous work on active amino acid-type precursors[” has been added; and in lines 289-290, the sentence “The enhanced HAA and/or HAL formation has been previously observed in HO-based AOP systems[44-45]” has been added.

[44] Y. Xiang, M. Gonsior, P. Schmitt-Kopplin, C. Shang, Influence of the UV/H2O2 advanced oxidation process on dissolved organic matter and the connection between elemental composition and disinfection byproduct formation[J]. Environmental Science & Technology, 2020, 54,(23): 14964-14973.

[45] J. Li, Z. Zhang, Y. Xiang, J. Jiang, R. Yin, Role of UV-based advanced oxidation processes on NOM alteration and DBP formation in drinking water treatment: A state-of-the-art review[J]. Chemosphere, 2023, 311, 136870.

Reviewer 2 Report

Specific Comments:

Line 35 (and elsewhere) – it is better to use the expression UV/Chlorine, since Cl2 is not involved in the process.

Line 35 and the rest of this paragraph – It would be helpful to cite at least one of the recent reviews of the UV/chlorine process, such as J. De Laat and M. I. Stefan (2018), UV/chlorine process in M.I. Stefan, Ed., Advanced Oxidation Processes for Water Treatment, IWA Publishing, London, UK, Chapter 9.

Line 44 – references 4 and 12 and references 5 and 7 are duplicates.

Line 92 – Figure S1 should be in the main text. Figure S1 should have dimensions.

Reference 18 is not cited correctly.

Line 93 (and elsewhere) – the correct term is irradiance; where was the irradiance measured? How was the spectrophotometer calibrated? Please explain how the irradiance was calculated from the spectral irradiance values outputted by the spectrophotometer (this is not explained in ref. 18).

Line 116 – was this chlorine concentration the concentration of Cl2 or the concentration of active chlorine (as HOCl + OCl)?

Table S1 – this table should be in the main text; what were the errors?

Tables 2 and 3 – again what were the errors?

Figures 1–5 are very confusing. What concentration is being displayed? The coding should be explained in the captions.

Line 232 (and elsewhere) – are these values really known to 5 significant figures?

Figure 6 – what are the units of the x and y labels?

General Comments:

This is an interesting paper; however, the authors need to do a better job of explaining their results. It is distressing that there is no consideration of errors or error analysis.

Author Response

General Comments:
This is an interesting paper; however, the authors need to do a better job of explaining their results. It is distressing that there is no consideration of errors or error analysis

Response: Thanks for pointing out the mistake. The error bars have been added to all the tables.

Specific Comments:
1. Line 35 (and elsewhere) – it is better to use the expression UV/Chlorine, since Cl2 is not involved in the process.

Response: Thanks for your suggestion. Cl2 was consistently replaced by chlorine in the whole manuscript.
2. Line 35 and the rest of this paragraph – It would be helpful to cite at least one of the recent reviews of the UV/chlorine process, such as J. De Laat and M. I. Stefan (2018), UV/chlorine process in M.I. Stefan, Ed., Advanced Oxidation Processes for Water Treatment, IWA Publishing, London, UK, Chapter 9.
Response: Thanks for your suggestion. As suggested, the recommended reference was added to line 37. Moreover, several more references were added to this paragraph. The references added can refer to Comment 1’s response for Reviewer 1.
3. Line 44 – references 4 and 12 and references 5 and 7 are duplicates.
Response: We apologize for the mistake. Now references 7 and references 12 have been deleted.
4. Line 92 – Figure S1 should be in the main text. Figure S1 should have dimensions.

Response: As requested, the dimensions of Figure S1 have been added, and the revised version has been moved to the main manuscript and numbered as Fig. 1. The revised Figure 1 now showed:

Fig. 1 Schematic diagram of the ultraviolet collimated beam apparatus

  1. Reference 18 is not cited correctly.
    Response: Thanks for pointing out the mistake. We have made corrections and the revised reference now read:

[25] Bolton J R, Linden K G. Standardization of methods for fluence (UV dose) determination in bench-scale UV experiments. Journal of Environmental Engineering[J]. 2003;129:209-15.

  1. Line 93 (and elsewhere) – the correct term is irradiance; where was the irradiance measured? How was the spectrophotometer calibrated? Please explain how the irradiance was calculated from the spectral irradiance values outputted by the spectrophotometer (this is not explained in ref. 18).

Response: Thanks for pointing out the mistake. The incident irradiance at the water surface was measured as 10 mW/cm2 using a calibrated Ocean Optics spectrophotometer (USB4000). The Ocean Optics spectroradiometer (USB4000) was calibrated by the manufacturer before use. The radiometer was used to acquire the lamp’s spectrum and intensity at different wavelengths from 200 nm to 400 nm. After obtaining the relative spectral emittance of the MP lamp (shown in Fig. S1), the fluence rate from 200 to 400 nm was calculated proportionally.

The description in lines 95-98 was revised and now read: The average irradiance (10 mW/cm2) was obtained using the calibrated Ocean Optics Spectro radiometer (USB4000), and the UV dose (600 mJ/cm2, a typical dose used practically)[25] was obtained by multiplying the average irradiation intensity with the irradiation time.
7. Line 116 – was this chlorine concentration the concentration of Cl2 or the concentration of active chlorine (as HOCl + OCl)?
Response: The chlorine concentration measured represented the concentration of total active chlorine (as HOCl + OCl) in the system. This point has been clarified in line 116:

Chlorine demand (total active chlorine) and DBP yields (i.e. DBPFP) were analyzed after 24 h post-chlorination.

  1. Table S1 – this table should be in the main text; what were the errors?
    Response: As suggested, we have put Table S1 in the main text and listed as Table 2 and the error bars of the data have been added.

Table 2. Chlorine decay during 60 s pre-oxidation of the seven model compounds under conditions: chlorine dose = 10 mg Cl2/L, pH = 6.0 or 8.0, [model benzene precursors]0 = 3 mg-C/L, UV fluence rate = 10 mW/cm2, [Br-] = 1 mg/L.

Model precursors

pH 6 (mg Cl/mg C)

pH 8 (mg Cl/mg C)

Ambient

Bromide-spiked

Ambient

Bromide-spiked

Cl2 alone

UV/Chlorine

Cl2 alone

UV/Chlorine

Cl2 alone

UV/Chlorine

Cl2 alone

UV/Chlorine

BA

0.05±0.0

0.37±0.0

0.01±0.0

0.38±0.0

0.02±0.0

0.96±0.1

0.01±0.0

0.98±0.1

NB

0.01±0.0

0.33±0.0

0.01±0.0

0.30±0.0

0.01±0.0

0.63±0.0

0.01±0.0

0.66±0.0

PN

0.13±0.0

0.93±0.1

0.12±0.0

0.98±0.1

0.2±0.0

1.34±0.1

0.24±0.0

1.47±0.2

RSC

3.34±0.3

3.38±0.3

3.30±0.3

3.37±0.3

3.30±0.2

3.37±0.3

3.30±0.3

3.37±0.4

2-MCP

0.23±0.0

1.07±0.0

0.53±0.0

1.37±0.0

0.27±0.0

1.53±0.1

0.55±0.0

1.65±0.1

4-MCP

0.10±0.0

1.33±0.1

0.08±0.0

1.57±0.1

0.13±0.0

1.73±0.1

0.17±0.0

1.83±0.1

2,4,6-TCP

0.20±0.0

0.70±0.0

0.23±0.0

1.37±0.1

0.17±0.0

1.15±0.1

0.21±0.0

1.37±0.1

  1. Tables 2 and 3 – again what were the errors?

Response: As suggested, the error bars of the data have been added in tables 2 and 3.

  1. Figures 1–5 are very confusing. What concentration is being displayed? The coding should be explained in the captions.
    Response: As suggested, the specific DBPs showed in the 5 figures were supplemented in the caption, and the conditions have been also described in the revised captions.
    11. Line 232 (and elsewhere) – are these values really known to 5 significant figures?

Response: Those values in line 232 and other places were checked and confirmed to be included in the corresponding figures.
12. Figure 6 – what are the units of the x and y labels?
Response: The units of the x and y labels are mg chlorine/mg C and µg/mg C, respectively, which have been added to the figure.

Round 2

Reviewer 1 Report

The authors have addressed the comments. Paper can be accepted.

Reviewer 2 Report

The authors have adequately addressed the reviewers comments